# Common Challenges and Solutions Associated with the Preparation of Silicone-Injected Human Head and Neck Vessels for Anatomical Study

**DOI:** 10.3390/brainsci11010032

**Published:** 2020-12-31

**Authors:** Musa Çırak, Kaan Yağmurlu, Sauson Soldozy, Pedro Norat, Mark E. Shaffrey, Mohammad Yashar S. Kalani

**Affiliations:** 1Department of Neurological Surgery, University of Virginia, Charlottesville, VA 22903, USA; musacirak@hotmail.com (M.Ç.); SS2AH@hscmail.mcc.virginia.edu (S.S.); pedronorat@hotmail.com (P.N.); mes8c@hscmail.mcc.virginia.edu (M.E.S.); 2Department of Neuroscience, University of Virginia, Charlottesville, VA 22903, USA; 3Department of Neurosurgery, St. John’s Neuroscience Institute, School of Medicine, University of Oklahoma, Tulsa, OK 74104, USA; stemcelldoctor@gmail.com

**Keywords:** human brain, cerebral vessels, head, microsurgery, neuroanatomy, neurosurgery, silicone injection methods

## Abstract

Neuroanatomy laboratory training is crucial for the education of neurosurgery residents and medical students. Since the brain is a complex and three-dimensional structure, it is challenging to understand the anatomical relationship of the cortex, internal structures, arteries, and veins without appropriate adjuncts. Several injection agents—including the inks/dyes, latex, polyester, acrylic resins, phenol, polyethylene glycol, and phenoxyethanol—have been explored. Colored silicon injection protocols for the head and neck vessels’ perfusion have greatly aided the study of neuroanatomy and surgical planning. This report presents a colored silicone injection method in detail, and also highlights the technical shortcomings of the standard techniques and workarounds for common challenges during 35 human cadaveric head injections. The human cadaveric head preparation and the coloring of the head vessels are divided into decapitation, tissue fixation with 10% formalin, the placement of the Silastic tubing into the parent vessels, the cleaning of the vessels from clots, and the injection of the colored silicone into the vessels. We describe the technical details of the preparation, injection, and preservation of cadaveric heads, and outline common challenges during colored silicone injection, which include the dislocation of the Silastic tubing during the injection, the injection of the wrong or inappropriate colored silicone into the vessel, intracranial vessel perforation, the incomplete silicone casting of the vessel, and silicone leakage from small vessels in the neck. Solutions to these common challenges are provided. Ethyl alcohol fixed, colored human heads provided the long-term preservation of tissue, and improve the sample consistency and preservation for the teaching of neuroanatomy and surgical technique.

## 1. Introduction

In the postmortem period, the brain is the most complicated organ in the body to preserve for long-term use. Part of the challenge with preserving the brain is its encasement by the skull and meninges, which limit deep preservation. Furthermore, the cerebral vasculature is fragile and subject to disruption while handling. Properly-preserved brains are necessary for training students in neuroanatomy and surgical techniques, and for postmortem analysis for medico-legal purposes [1]. In addition to appropriate preservation, adequate coloration is essential in order to enable differentiation of microvascular structures. Many groups have experimented with different fixation and injection techniques in order to produce high-quality heads and brains for teaching and training [2,3,4,5].

The very first description of the use of injection methods to visualize neurovascular structures was by Thomas Willis and his colleague Christopher Wren in 1684, allowing them to characterize the ‘circle of Willis’ in humans correctly [6,7]. Fast forward several centuries, several injection agents have since been explored. This includes the use of inks/dyes, latex, polyester, acrylic resins, phenol, polyethylene glycol, and phenoxyethanol [5,8]. Past challenges include difficult preparation and high viscosity, prolonging overall injection time and limiting the vascular penetration into the small distal vessels. Specific agents are also more prone to leakage. Besides this, the achievement of the appropriate hardness without becoming too brittle or sacrificing elasticity has been challenging.

This study reports a simple, reliable preservation and injection technique for tissue preservation, and discusses the overcoming of common shortcomings with this technique.

## 2. Material and Methods

We injected silicone into 35 human cadaveric heads between the years of 2012–2020. Before the decapitation, the carotid artery and internal jugular vein are dissected on one side of the neck in order to inject tap water, for the removal of the clots, and 10% formalin solution (Neutral buffered formalin, ThermoScientific^®^, Newark, DE, USA), for brain tissue fixation through the vessels. This procedure takes two hours at the anatomical board. Our recipe includes decapitation and tissue fixation, the tubing of the parent vessels, washing out the vessels, and the preparation and injection of silicone sequentially.

### 2.1. Decapitation and Tissue Fixation with Solution

At the seventh cervical vertebra level, the head is cut cleanly using a bone saw and knife for soft tissues [4]. A decent cut is essential for the isolation and cannulation of the parent vessels, and the clamping of the small neck vessels that we discuss later on. After decapitation, the head is fixed with a 10% formalin solution (Neutral buffered formalin, Thermo scientific) for at least one month. Before starting the dissection and injection of the vessels, the fixative solution is changed to 70% ethyl alcohol (200 proof pure ethanol, Decon^®^, King of Prussia, PA, USA) due to the formaldehyde’s carcinogenic effect [9].

### 2.2. The Tubing of the Parent Arteries and Veins

The common carotid arteries, vertebral arteries, and internal jugular veins are meticulously dissected away from the soft tissues, and a Tygon Silastic tubing (Fisher Scientific, Pittsburgh, PA, USA) is placed inside each vessel and fixed with sutures (2.0 or 3.0 silk) on both sides (Figure 1A).

### 2.3. Washing Out the Vessels

Large (P1000) disposable polypropylene pipettes (Fischer Scientific) are placed into the tubing placed in each vessel. Each vessel is cleaned from coagulated blood and serum by the administration of tap water with a 60 mL syringe (320 mL for each vessel daily) for five days, three times a day. During this procedure, the water should not be injected with high pressure, in order to avoid vessels bursting and disruption. Clamps prevent the leakage from the small neck vessels. Epidural and subdural spaces are tamponed by paper and bone wax in order to prevent leakage.

After five days of washing, all of the vessels should be clean. If residual clots continue to be drained from the vessels, the cleaning should be extended for 2–3 additional days. Between the procedures, the head is stored inside the 70% ethyl alcohol solution.

### 2.4. Preparation and Injection of the Silicone

Silicone (3110 RTV silicone rubber; Dow Corning, Midland, MI, USA) and thinner (polydimethylsiloxane: 200 fluid 20 CST; Dow Corning) are mixed, for arteries, thinner:silicone 2:1; and for veins, thinner:silicone 1:1 (Figure 1B–D). After that, powder paint (red for arteries and blue for veins) (Crayola) is added until a vivid color is obtained. For the injection of the carotid arteries, a total of 40 mL mixtures including paint, silicone and thinner, is necessary. The vertebral arteries require 20 mL for proper casting. For an internal jugular vein, 100 mL of the mixture is essential. Finally, one third of the catalyst tube is added for each 100–150 mL mixture of the silicone, thinner, and powder paint. The colored silicone injections start with the arteries first, and then the veins. The red-colored silicone is gently injected, and in the case of pressure build-up in the vessel, the procedure is stopped, and then it is started slowly again.

The perforator arteries are more challenging to perfuse; the key is to inject slowly with low pressure. In prolonged injections, the catalyst makes the silicone solidify relatively quickly, which is not optimal for the perfusion of small-sized vessels. Conversely, a faster injection may burst the vessel wall and leak the silicone into the subarachnoid space in the intracranial area. During the injection, a sudden drop in the vessel’s pressure means that the vessel has burst. The injection should be performed slowly in order to minimize the injection into the brain parenchyma. Before the completion of the first half of the silicon injection (the first 20 mL silicone injection) into the carotid artery, if the silicone comes out of the other carotid artery, the injection should be stopped and both carotid arteries should be clamped. The injection of the contralateral carotid artery can then be resumed without unclamping the previously-injected carotid artery. When high pressure is encountered, the injection should be stopped immediately. Once both carotid arteries are injected, the vertebral arteries should be injected with red-colored silicone in the same manner as the carotid arteries. Next, the internal jugular veins should be injected with blue-colored silicone in the same way. During the injection, the leakage from the small neck vessels can be blocked by clamping. Once four arteries and two veins are injected, the head is left in the sink for 24 h, taking care to avoid movement. After that, the clamps are removed, and the head is put into a fixation solution for two days in order to ready it for the dissection (Figure 2).

## 3. Results

### 3.1. Common Difficulties and Solutions Associated with Silicone Injection 

In our silicone injection experience on 35 cadaveric heads, we noted some common difficulties in order, from most common to least common (Table 1 and Figure 3).

### 3.2. Silicone Leakage from Small Vessels in the Neck

The vessels are clamped in order to prevent leakage. If this is not possible, the pressure is applied over the vessels. 

### 3.3. Poor Filling of Silicone in the Vessel

The silicone paint may not fill appropriately into the vessel if the amount of catalyst added is excessive. In cases of poor vessel filling by the color, the viscosity of the silicone should be checked. If the viscosity is appropriate, two additional factors may contribute to poor filling. First, the vessel may be obstructed by a clot or a foreign body, or, second, the entire vascular tree may already be cast. In the first situation, the injection is paused for 1–2 min, and then continued. In the second scenario, the injection of the vessel is terminated, and the injection of the contralateral side should be started.

### 3.4. Intracranial Vessel Perforation

As was mentioned previously in the method section, sudden pressure decline is caused by vessel perforation. In this case, the injection is maintained very slowly.

### 3.5. Dislocation of the Silastic Tubing During Injection

The dislocation of the Silastic tubing may occur when insufficient dissection of the vessel from the surrounding soft tissues has occurred. The tubing may also become dislodged if the suturing of the main vessels is inadequate. Another reason is that high pressure during the injection can dislocate the tubing from the vessel. The vessel should be clamped immediately in order to prevent the silicone from extruding from the vessel in this situation.

### 3.6. Injection of Wrong Color Silicone into the Vessel

Red-colored silicone can mistakenly be injected into the vein or vice versa. The injection should be aborted immediately, and the silicone should be removed from inside the vessel using a sucker, insofar as is possible. Deep aspiration of silicone from the vessel should be avoided due to the risk of vessel perforation. After the aspiration of colored silicone, the contralateral vessel should be carefully injected with the correct color in order to dislodge and deliver the incorrect paint from the inappropriately-injected vessel.

## 4. Discussion

The head and neck vessels can be divided into extracranial and intracranial vessels, which can be further subdivided into cortical and deep vessels (Figure 2A–F). During the silicone injection, the extracranial vessels fill first, followed by the intracranial vessels. The colored silicone fills the large vessels first, and then proceeds to the small vessels based on the physic rule, in which the small vessels have a higher resistance than large vessels. Deeper vascular structures can also be visualized without leakage, indicating the adequate perfusion of all of the cranial vessels by this methodology. Figure 2D demonstrates the dissection of the posterior fossa vasculature, including the sinuses (superior petrosal sinus, sigmoid sinus) and arterial vessels (basilar artery, anterior inferior cerebellar artery, vertebral arteries, and anterior spinal artery). In Figure 2E,F, we provide dissections of deeper intracranial structures and their adjacent vessels. We show that the lenticulostriate arteries, despite their size and location, received adequate silicone injection, demonstrating that deep microvasculature penetration is feasible, with excellent results.

Although the microsurgical dissection at the neuroanatomy laboratory is time-consuming, it is the best way to learn the complex three-dimensional architecture of the cerebrum and skull base. Besides the educational gains, the neuroanatomy lab training also provides an opportunity for the discovery of new anatomical variations that may be significant in disease states or for treatment [10,11,12,13,14,15,16] (Figure 2).

The use of formaldehyde is limited in current practice due to its carcinogenic effects [9]. The head should be kept in formaldehyde or formalin for the initial fix, but should be switched to 70% ethyl alcohol for long-term preservation. The ethyl alcohol does not change or destroy the color or stiffness of the skin, muscle, bone, and brain tissue significantly, as the formaldehyde does. Since the cost of methyl alcohol is cheaper, some centers are using it instead of ethyl alcohol. Methyl alcohol has a lower fixation effect than ethyl alcohol, which causes a softer brain tissue than may be expected for a hands-on course. Still, it is not proper for microsurgical dissection for research purposes, because the hard brain tissue allows for retraction or manipulation without destroying the tissue.

Other injection materials—including inks/dyes, latex, polyester, acrylic resins, and polyurethane—have been used, depending on the dissection’s goal [8,17]. Inks and dyes stain the vessels, with a high percentage of leakage [17]. Latex and polyester have a minimal corrosive resistance, and acrylic resins are rigid and brittle materials that are used for vascular corrosion casting projects [8]. Polyurethane elastomer was described as an alternative to silicone with advantages in its desired hardness, sufficient elasticity, and high tear resistance [8,17,18]. One study compared the polyurethane elastomer’s effectiveness in latissimus dorsi muscle vascularization [8]. It concluded that polyurethane elastomer is useful for the study of the experimental development of new flaps in plastic surgery, including perforator flaps with a long course of tiny vessels [8,19]. Based on elasticity, silicone rubber is still the favorite injection material for the microsurgical dissection of the vessel, especially for neurovascular structures [4,5]. For decades, silicone rubber has been the most commonly-used material for cranial microsurgical studies, with excellent outcomes [17,20].

Dr. Albert L. Rhoton made outstanding contributions to microsurgical neuroanatomy by publishing articles and books, and by training over 100 fellows worldwide [21,22]. His tissue fixation and colored silicone injection techniques are still widely used [4]. At the Skull Base Laboratory at the University of Virginia, we continue to use a modification of his recipe. Although Dr. Rhoton’s laboratory graduates are familiar with the troubleshooting techniques presented, these techniques have not been previously published. In this study, we reported strategies to overcome difficulties during the procedure in detail, and in a stepwise manner.

## 5. Conclusions

The study of cadaveric neuroanatomy is crucial for education and research. Ethyl alcohol fixed, colored silicone injected human heads provide an excellent means for practical training and the long-term preservation of tissue.

## Figures and Tables

**Figure 1 brainsci-11-00032-f001:**
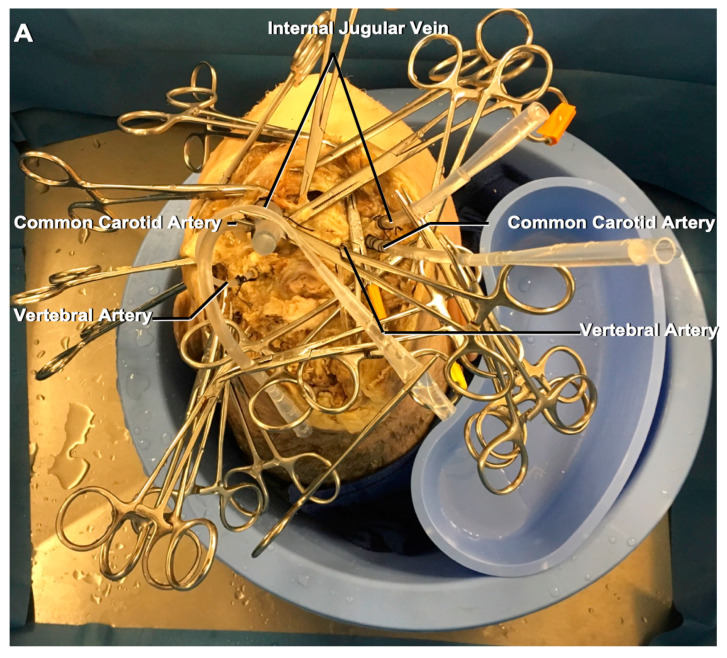
(**A**) The placement and fixation of the Silastic tubing into the carotid and vertebral arteries and internal jugular veins in order to prepare the vessels for cleaning and silicone injection on both sides. The small vessels of the face and neck are clamped in order to prevent the leakage of the paint. (**B**) Table 1 preparation before starting the injection procedure. The head is placed in the sink, and the required materials are placed on the sides. (**C**) The silicone, powder paint, catalyst, thinner, and the different sized clamps. (**D**) Different sized syringes (60 mL, 10 mL, and 5 mL) and pipettes. A new syringe should be used for each vessel because the catalyst causes the silicone’s solidification.

**Figure 2 brainsci-11-00032-f002:**
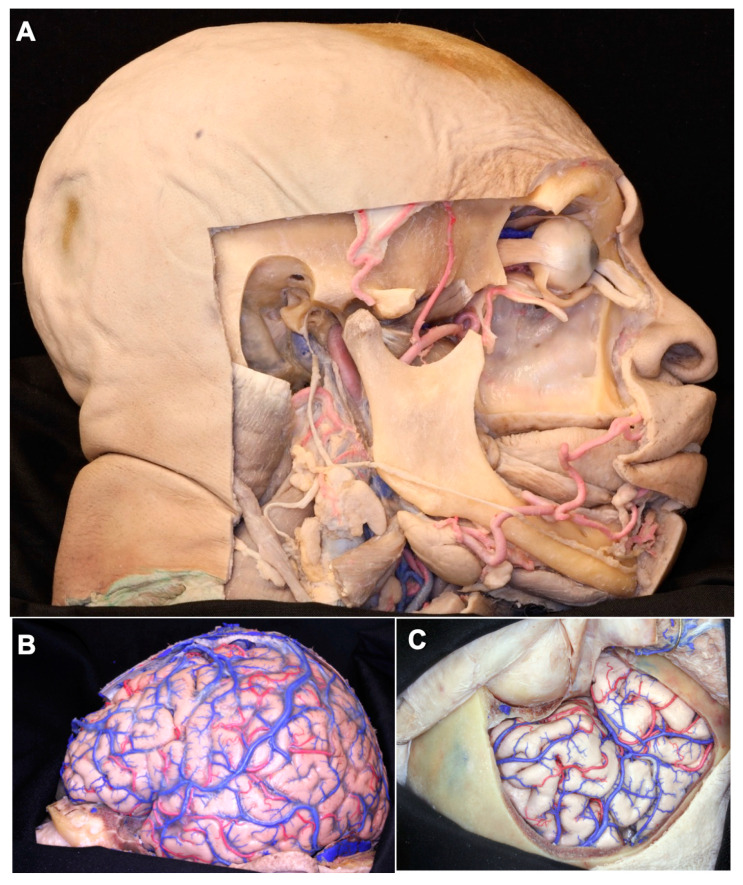
(**A**) Usage of the proper fixation solution provides a tissue fixation with a real color of bone, muscle, and fat tissue. Well-injected extracranial arteries and veins are exposed. (**B**) The cortical arteries, veins, and sinuses were very well injected. (**C**) The cortical arteries and veins are seen after the removal of the arachnoid membrane. A suitable tissue fixation method provides a stiff and less-fragile brain tissue, which eases the microdissection. (**D**) The posterior fossa vessels and brainstem. (**E**) The appearance, after silicone injection, of the deep arteries and veins. (**F**) The silicone-injected small branches of the anterior and middle cerebral arteries are exposed.

**Figure 3 brainsci-11-00032-f003:**
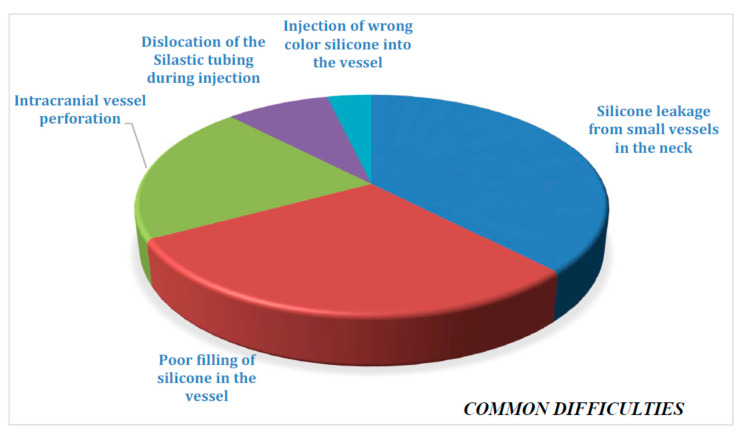
Common difficulties.

**Table 1 brainsci-11-00032-t001:** Number and ratios of common difficulties.

Common Difficulties	Of 35 Heads	Ratio of Frequency
Silicone leakage from small vessels in the neck	31	88.5%
Poor filling of silicone in the vessel	24	68.5%
Intracranial vessel perforation	17	48.5%
Dislocation of the Silastic tubing during injection	7	20%
Injection of wrong color silicone into the vessel	3	8.5%

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
