# Peer review of "Common Challenges and Solutions Associated with the Preparation of Silicone-Injected Human Head and Neck Vessels for Anatomical Study"

_brainsci, 2020, doi:10.3390/brainsci11010032_

Round 1
Reviewer 1 Report
In the article “Common Challenges and Solutions Associated with Preparation of Silicone Injected Human Head and Neck Vessels for Anatomical Study”, the authors Musa Çırak, Kaan YaÄŸmurlu, Sauson Soldozy, Pedro Norat, Mark E. Shaffrey, M. Yashar S. Kalani, describe a protocol for the injection of the head and neck vessels based in coloured silicon.
A multitude of various materials have been employed for the visualization of cadaveric vessels: gelatin, latex, Indian ink, acrylates, polyurethanes and a long etcetera, including the silicone employed by the authors themselves. Actually, silicone casting has been used for decades. Therefore, my main concern is that the authors do not clearly explain what is new about their methodology compared to the techniques available today.
Moreover, they should discuss and weigh the advantages and disadvantages of their method against other materials (polyurethanes, for instance) and other methods currently employed
Although the main drawbacks of their methodology are quantified -by the way, with a high percentage of problematic perfusions- aspects such as viscosity, the shrinkage, time of hardening, elasticity and resistance obtained should be evaluated in comparison to the other techniques available. For instance, how does the authors methodology improve the hardness, elasticity and tear resistance obtained with the polyurethane perfussion?
Additionally, I have certain reservations about the realization of the photographs. The quality of them is very good, but the images do not seem real. There are no imperfections in the perfusion and the tissues lack connective tissue, adipose, etc that gives them an artificial look. I don't dispute the clarity or impact of the images, but raw images would be more appropriate to assess the quality of the perfusion.
The writing is sloppy with grammatical errors and typos. For instance, in the sentence: “The silicone filling in the smaller vessels in diameter more difficult than in the large vessel due to ...” Verbs is absent and large vessel should be large vessels.
The format of the bibliographic citations should be standardized. The exact date of publication of the article is not relevant and is not shown in all cases.
All the indications to the references in the text are written after the point in the style of “training. [2-5]”. The dot should be located after the bracket. The same happens with the reference to the figures in the text.
Table 1 and Graph 1 are redundant. They should be grouped.
Regarding the methodology, It would be relevant to know the time elapsed from the death to the fixation of the heads. Also, the question arises of how after one month in immersion in formalin it is achieved the adequate penetration of the fixative to the deepest structures. Have the authors considered employing the Thiel solution instead formalin?.
In abstract Line 14 “There have been many studies on the preservation of brain tissue.” This is a quite obvious observation, that does not enlighten the abstract.
Author Response
Reviewer 1.
- In the article “Common Challenges and Solutions Associated with Preparation of Silicone Injected Human Head and Neck Vessels for Anatomical Study”, the authors Musa Çırak, Kaan YaÄŸmurlu, Sauson Soldozy, Pedro Norat, Mark E. Shaffrey, M. Yashar S. Kalani, describe a protocol for the injection of the head and neck vessels based in coloured silicon.
Respond #: First of all, we appreciate the reviewer for his/her professional and detail comments, which makes our paper stronger, we believe.
- A multitude of various materials have been employed for the visualization of cadaveric vessels: gelatin, latex, Indian ink, acrylates, polyurethanes and a long etcetera, including the silicone employed by the authors themselves. Actually, silicone casting has been used for decades. Therefore, my main concern is that the authors do not clearly explain what is new about their methodology compared to the techniques available today. Moreover, they should discuss and weigh the advantages and disadvantages of their method against other materials (polyurethanes, for instance) and other methods currently employed. Although the main drawbacks of their methodology are quantified -by the way, with a high percentage of problematic perfusions- aspects such as viscosity, the shrinkage, time of hardening, elasticity and resistance obtained should be evaluated in comparison to the other techniques available. For instance, how does the authors methodology improve the hardness, elasticity and tear resistance obtained with the polyurethane perfussion?
Respond #: This paper aims to describe the technical challenges and overcome nuances in silicone injection instead of describing a new method. Based on those unpublished subjects, we would say that this original paper is a unique contribution to the literature. The comparison of the injection materials could be another project. As we noted in the last paragraph of the discussion section, the silicone injection method has already been published before (please see reference 4), but the common challenging and handling techniques performing by K.Y. (one of the former fellow of Dr. Rhoton) not published before. That’s is the reason why we decide to write this paper. We are pleased to use the silicone rubber due to satisfactory results (Please check some comments and neuroanatomy publications used silicone injection based on our recipe below). We would appreciate it if the reviewer let us focus on our experience with the silicone injection in this paper.
- Rhoton Jr AL. Rhoton's cranial anatomy and surgical approaches. Oxford University Press; 2019.
- Spetzler RF, Kalani MY, Nakaji P, Yagmurlu K. Color atlas of brainstem surgery. Thieme; 2017.
- Awad IA. Book Review: Color Atlas of Brainstem Surgery.
- Sorenson J, Khan N, Couldwell W, Robertson J. The Rhoton collection. World Neurosurgery. 2016;92:649-52.
- Matsushima T, Matsushima K, Kobayashi S, Lister JR, Morcos JJ. The microneurosurgical anatomy legacy of Albert L. Rhoton Jr., MD: an analysis of transition and evolution over 50 years. Journal of neurosurgery. 2018;129(5):1331-41.
- Kalani MY, Yagmurlu K, Martirosyan NL, Cavalcanti DD, Spetzler RF. Approach selection for intrinsic brainstem pathologies. Journal of Neurosurgery. 2016;125(6):1596-607.
- Yagmurlu K, Vlasak AL, Rhoton Jr AL. Three-dimensional topographic fiber tract anatomy of the cerebrum. Operative Neurosurgery. 2015;11(2):274-305.
- Yagmurlu K, Rhoton Jr AL. Lateral and third ventricle anatomy. Neuroendoscopic Surgery. 2016.
- YaÄŸmurlu K, Sokolowski JD, Çırak M, Urgun K, Soldozy S, Mut M, Shaffrey ME, Tvrdik P, Kalani MY. Anatomical Features of the Deep Cervical Lymphatic System and Intrajugular Lymphatic Vessels in Humans. Brain Sciences. 2020;10(12):953.
Based on the reviewer’s comment, we have added one paragraph with new citations about other materials in the discussion section on page 9: “Other injection materials, including ink/dyes, latex, polyester, acrylic resins, and polyurethane, have been used depending on the dissection's goal [8,17]. Ink and dyes stain the vessels with a high percentage of leakage [17]. Latex and polyester have a minimal corrosive resistance, and the acrylic resins is rigid and brittle material used for vascular corrosion casting projects [8]. Polyurethane elastomer was described as an alternative to silicone with advantages of the desired hardness, sufficient elasticity, high tear resistance [8,17,18]. One study compared the polyurethane elastomer's effectiveness in latissimus dorsi muscle vascularization [8]. It concluded that polyurethane elastomer is useful for studying the experimental development of new flaps in plastic surgery, including perforator flaps with a long course of tiny vessels [8,19]. Based on elasticity, silicone rubber is still the favorite injection material for microsurgical dissection of the vessel, especially for neurovascular structures [4,7]. For decades, silicone rubber has been the most commonly used material for cranial microsurgical studies with excellent outcomes [17,20].”
New citations as follows:
- Salma A, Chow A, Ammirati M. Setting up a microneurosurgical skull base lab: technical and operational considerations. Neurosurgical review. 2011;34(3):317-26.
- Heymans OY, Nelissen XP, Peters S, Lemaire V, Carlier A. New approach to vascular injection in fresh cadaver dissection. J Reconstr Microsurg 2004;20:311–315.
- Rhoton Jr AL. Rhoton's cranial anatomy and surgical approaches. Oxford University Press; 2019.
- Additionally, I have certain reservations about the realization of the photographs. The quality of them is very good, but the images do not seem real. There are no imperfections in the perfusion and the tissues lack connective tissue, adipose, etc that gives them an artificial look. I don't dispute the clarity or impact of the images, but raw images would be more appropriate to assess the quality of the perfusion.
Respond #: We take this comment as a compliment because all pictures are real and raw, and no photoshop has been used. What makes the dissection pictures excellent is our recipe for silicone injection method, as noted in this paper, photographing with HD quality and using a flash ring and macro lenses; for details, please see reference 4. Those pictures are called Rhotonish style. If the reviewer wants to see more dissection pictures prepared with similar injection and photograph methods, please look at Dr. Rhoton‘s book.
Rhoton Jr AL. Rhoton's cranial anatomy and surgical approaches. Oxford University Press; 2019 Oct 3.
- The writing is sloppy with grammatical errors and typos. For instance, in the sentence: “The silicone filling in the smaller vessels in diameter more difficult than in the large vessel due to ...” Verbs is absent and large vessel should be large vessels.
Respond #: Thanks for pointing the syntax error out. We have deleted that confusing sentence and modified the next sentence in the first paragraph of the discussion section on page 9: “The colored silicone fills the large vessels first, and then proceed the small vessels based on the physic rule, which the small vessels have a higher resistance than large vessels.” We have also gone through the manuscript to correct grammatical errors. All changes can be visible with “track changes.”
- The format of the bibliographic citations should be standardized. The exact date of publication of the article is not relevant and is not shown in all cases.
Respond #: We have removed the dates and months and left only years in the citations. Thanks!
- All the indications to the references in the text are written after the point in the style of “training. [2-5]”. The dot should be located after the bracket. The same happens with the reference to the figures in the text.
Respond #: We have placed all dots after the bracket and figure citations in the text. Thanks!
- Table 1 and Graph 1 are redundant. They should be grouped.
Respond #: Graph 1 is the graphical version of table 1. We did put the graph to make the reading of table 1 easier for readers. Graph 1 can be removed if the editor agrees with the reviewer. Thanks!
- Regarding the methodology, It would be relevant to know the time elapsed from the death to the fixation of the heads. Also, the question arises of how after one month in immersion in formalin it is achieved the adequate penetration of the fixative to the deepest structures. Have the authors considered employing the Thiel solution instead formalin?.
Respond #: K.Y., one of the co-authors, studied the cadavers embalming with the Thiel method, and the result was unsatisfactory. The Thiel solution is a soft-fixed and useful method for the visceral organs and large muscle groups to keep the tissue fresh, but it is not proper for the brain tissue, making it too much softer (like a soup viscosity), which is not dissectible. The formalin-based solutions are already well known as the hard-fixed embalming. For further information about formalin, please check the article below.
Balta JY, Cronin M, Cryan JF, O'MAHONY SM. Human preservation techniques in anatomy: A 21st-century medical education perspective. Clinical Anatomy. 2015 Sep;28(6):725-34.
- In abstract Line 14 “There have been many studies on the preservation of brain tissue.” This is a quite obvious observation, that does not enlighten the abstract.
Respond #: We have removed that sentence and added, “Several injection agents including the inks/dyes, latex, polyester, acrylic resins, phenol, polyethylene glycol, and phenoxyethanol have since been explored.” to the abstract section.

Reviewer 2 Report
It is an excellent study of great practical value. The article submitted for my review fills a severe gap in anatomical preparation, especially regarding advanced brain vascular anatomy visualization techniques. The authors show, step by step, how to prepare the specimens for both scientific work and teaching purposes. The excellent and realistic illustration material is a strong side of the paper. Samples shown in the work are of the highest quality. It is an awe-inspiring study. I applied corrosion casting methods in my research, and I realize how complicated the procedure may be. Authors provide a detailed description of the preparation, injection, and preservation of cadaveric heads and outline common challenges during colored silicone injection.
Minor corrections
I suggest a small change of keywords: Human Brain; Cerebral Vessels; Head; Microsurgery; Neuroanatomy; Neurosurgery; Silicone Injection Methods.
Page 2, line 56 – "At the level of the seventh cervical spine" should be replaced by "At the seventh cervical vertebra level."
Paragraph" Decapitation and tissue fixation with a solution" should be clarified. It should be stressed whether the head was detached before fixation (from the fresh corpse)?
Page 5 – line 77/78 –" Each vessel is cleaned from coagulated blood and serum by administering tap 77 water with a 60 mL syringe (daily 320 mL for each vessel daily) for five days, three times a day". I would like to know if this procedure was performed after formaldehyde fixation? On page 2 (line 58/59), you stated, "After decapitation, the head is placed in 58 a 10% formalin solution (Neutral buffered formalin, Thermo scientific) for at least one month". Please, explain why preliminary cleaning of the blood vessels was not performed before formalin fixation? Formalin causes coagulation of blood, and cumulated blood cloths may be challenging to remove after fixation.
Page 5, line 94/95 – "Colored silicone injections are started with arteries first, and then veins." It would be valuable to know when to stop the arterial injection and start the infusion of veins. This clarification may be added after line 109 (page 5) – after the sentence "Next, the internal jugular veins should be injected with blue-colored in the same manner."
In Figure 2 description, changes should be introduced as follows "2E, after silicone injection of the deep arteries and veins" should be changed into "2E, appearance after silicone injection of the deep arteries and veins".
Author Response
- It is an excellent study of great practical value. The article submitted for my review fills a severe gap in anatomical preparation, especially regarding advanced brain vascular anatomy visualization techniques. The authors show, step by step, how to prepare the specimens for both scientific work and teaching purposes. The excellent and realistic illustration material is a strong side of the paper. Samples shown in the work are of the highest quality. It is an awe-inspiring study. I applied corrosion casting methods in my research, and I realize how complicated the procedure may be. Authors provide a detailed description of the preparation, injection, and preservation of cadaveric heads and outline common challenges during colored silicone injection.
Respond #: We appreciate the reviewer for his/her kind comments and corrections.
- Minor corrections
- I suggest a small change of keywords: Human Brain; Cerebral Vessels; Head; Microsurgery; Neuroanatomy; Neurosurgery; Silicone Injection Methods.
Respond #: We have changed the keywords as mentioned above. Thanks!
- Page 2, line 56 – "At the level of the seventh cervical spine" should be replaced by "At the seventh cervical vertebra level."
Respond #: We have changed it.
- Paragraph" Decapitation and tissue fixation with a solution" should be clarified. It should be stressed whether the head was detached before fixation (from the fresh corpse)?
Respond #: To make this issue clearer, we have made some changes as follows:
- The first paragraph of the material method section on page 2: “Before the decapitation, the carotid artery and internal jugular vein are dissected on the one side of the neck to inject tap water for removing the clots and the 10% formalin solution (Neutral buffered formalin, Thermo scientific) for brain tissue fixation through the vessels. This procedure takes two hours at the anatomical board.”
- ‘Decapitation and tissue fixation with solution’ part in page 2: “After decapitation, the head is fixed with a 10% formalin solution (Neutral buffered formalin, Thermo scientific) for at least one month.”
- Page 5 – line 77/78 –" Each vessel is cleaned from coagulated blood and serum by administering tap 77 water with a 60 mL syringe (daily 320 mL for each vessel daily) for five days, three times a day". I would like to know if this procedure was performed after formaldehyde fixation? On page 2 (line 58/59), you stated, "After decapitation, the head is placed in 58 a 10% formalin solution (Neutral buffered formalin, Thermo scientific) for at least one month". Please, explain why preliminary cleaning of the blood vessels was not performed before formalin fixation? Formalin causes coagulation of blood, and cumulated blood cloths may be challenging to remove after fixation.
Respond #: That’s a great point that will be nice to mention in the manuscript. To make it clear, we have added a sentence to the first paragraph of the material method section on page 2: “Before the decapitation, the carotid artery and internal jugular vein are dissected on the one side of the neck to inject tap water for removing the clots and the 10% formalin solution (Neutral buffered formalin, Thermo scientific) for brain tissue fixation through the vessels. This procedure takes two hours at the anatomical board.”
- Page 5, line 94/95 – "Colored silicone injections are started with arteries first, and then veins." It would be valuable to know when to stop the arterial injection and start the infusion of veins. This clarification may be added after line 109 (page 5) – after the sentence "Next, the internal jugular veins should be injected with blue-colored in the same manner."
Respond #: We added the sentence to the “preparation and injection of the silicone” on page 5: “Once both carotid arteries are injected, the vertebral arteries should be injected with red-colored silicone in the same manner as carotid arteries. Next, the internal jugular veins should be injected with blue-colored in the same way.”
- In Figure 2 description, changes should be introduced as follows "2E, after silicone injection of the deep arteries and veins" should be changed into "2E, appearance after silicone injection of the deep arteries and veins".
Respond #: We have changed it.

Round 2
Reviewer 1 Report
I have read the authors' replies to my comments and to those by the other reviewer. I am very glad to learn that the authors have addressed all the crucial comments in detail.
I would only add that in case of keeping Table 1 -in my opinion redundant with respect to the graph- the percent symbol in %88.5, should be placed correctly. Also the content of the cells should be centered.